# Enhanced Malignancy Prediction of Small Lung Nodules in Different Populations Using Transfer Learning on Low-Dose Computed Tomography

**DOI:** 10.3390/diagnostics15121460

**Published:** 2025-06-08

**Authors:** Jyun-Ru Chen, Kuei-Yuan Hou, Yung-Chen Wang, Sen-Ping Lin, Yuan-Heng Mo, Shih-Chieh Peng, Chia-Feng Lu

**Affiliations:** 1Department of Biomedical Imaging and Radiological Sciences, National Yang Ming Chiao Tung University, Taipei 112, Taiwan; r1c15j.be08@nycu.edu.tw; 2Department of Radiology, Cathay General Hospital, Taipei 106, Taiwan; circle@cgh.org.tw (K.-Y.H.); edwardwg@cgh.org.tw (Y.-C.W.); lsp@cgh.org.tw (S.-P.L.); cgh06058@cgh.org.tw (Y.-H.M.); cgh415823@cgh.org.tw (S.-C.P.); 3Department of Biomedical Imaging and Radiological Sciences, Chung Shan Medical University, Taichung 402, Taiwan; 4Department of Medicine, School of Medicine, Fu Jen Catholic University, Taipei 242, Taiwan

**Keywords:** small lung nodule, malignancy prediction, population variations, deep learning, transfer learning

## Abstract

**Background:** Predicting malignancy in small lung nodules (SLNs) across diverse populations is challenging due to significant demographic and clinical variations. This study investigates whether transfer learning (TL) can improve malignancy prediction for SLNs using low-dose computed tomography across datasets from different countries. **Methods:** We collected two datasets: an Asian dataset (669 SLNs from Cathay General Hospital, CGH, Taiwan) and an American dataset (600 SLNs from the National Lung Screening Trial, NLST, America). Initial U-Net models for malignancy prediction were trained on each dataset, followed by the application of TL to transfer model parameters across datasets. Model performance was evaluated using accuracy, specificity, sensitivity, and the area under the receiver operating characteristic curve (AUC). **Results:** Significant demographic differences (*p* < 0.001) were observed between the CGH and NLST datasets. Initial models trained on one dataset showed a substantial performance decline of 15.2% to 97.9% when applied to the other dataset. TL enhanced model performance across datasets by 21.1% to 159.5% (*p* < 0.001), achieving an accuracy of 0.86–0.91, sensitivity of 0.81–0.96, specificity of 0.89–0.92, and an AUC of 0.90–0.97. **Conclusions:** TL enhances SLN malignancy prediction models by addressing population variations and enabling their application across diverse international datasets.

## 1. Introduction

The 5-year survival rate of lung cancer patients could be improved by 30% by the patients receiving appropriate treatments at an early stage [1,2]. To achieve early-stage treatments, an effective screening approach for lung nodules is required. However, previous studies using tissue biopsy assessment have reported a more than 13% reduced diagnostic accuracy for small lung nodules (SLNs, less than 10 mm in diameter) compared to large lung nodules (with sizes between 10 and 30 mm) [3,4]. Considering the high prevalence [5] and low malignancy rate of SLNs [6], predicting the malignancy of SLNs is critical to reducing the waste of medical resources and significantly enhancing patient prognosis.

Deep-learning (DL) algorithms, including architectures like ResNet and U-Net, have demonstrated significant potential in extracting subtle image features from low-dose computed tomography (LDCT) to enhance malignancy prediction in SLNs [7,8]. These models leverage convolutional neural networks (CNNs) to capture complex patterns that may be imperceptible to human observers, thereby enhancing diagnostic accuracy in early lung cancer detection. However, despite these promising results, studies have shown a concern in terms of the poor international application for DL models [9,10], which indicated that DL models trained on data from one country often perform poorly when applied to datasets from different countries, reflecting the challenge of application across populations with varying demographic and clinical characteristics. For instance, Asian and American populations have inconsistent risk factors for lung cancer, including age at diagnosis, smoking history, and SLN types [11,12,13,14,15,16]. Accordingly, the application of malignancy prediction models based on the American dataset to the Asian dataset may have reduced performance, and vice versa. This limitation of international applications is not unique to lung cancer; similar issues have been reported in DL applications for other diseases, including chronic hepatitis B, coronary arterial calcifications, and mammographic lesions [17,18,19].

One fundamental barrier to improving international application is the difficulty of exchanging medical imaging data across countries due to privacy concerns and regulatory restrictions. Consequently, many DL models are developed using datasets from a single country or ethnic group, which limits their robustness when deployed in different clinical settings characterized by variations in imaging protocols and population genetics [20]. While pooling data from multiple regions could theoretically improve model generalizability, it also introduces heterogeneity that complicates training and may degrade performance [21]. Techniques such as intensity normalization and spatial smoothing have been employed to mitigate scanner-related variability [22], but these methods cannot fully account for population-level differences in genetics, diet, and lifestyle factors that influence disease presentation. To address these challenges, transfer learning (TL) has emerged as a promising strategy. TL involves reusing feature extraction layers from a model pretrained on one population and fine-tuning the classifier layers on data from a target population, thereby adapting the model to local characteristics without requiring extensive retraining from scratch [23]. We anticipated that TL could facilitate the international application of SLN malignancy prediction by enhancing the model performance across different populations.

In this study, we investigated the impact of population variations on the SLN malignancy prediction between American and Asian datasets and further proposed the TL technique to overcome this influence. This study was designed to achieve the following aims: first, to identify the differences in SLN characteristics between Asian and American datasets; second, to propose a malignancy prediction model for SLNs based on LDCT using a DL approach; third, to assess the reduction of prediction performance when applying models across two datasets; and finally, to evaluate the efficacy of TL in improving prediction models for SLN malignancy.

## 2. Materials and Methods

### 2.1. Study Cohort

This study built an Asian dataset of SLNs by retrospective collection of patients with lung nodules in Taipei, Hsinchu, and Sijhih branches of the Cathay General Hospital (CGH) system from 2006 to 2022. Afterward, we excluded patients with unavailable histology for malignancy, insufficient quality of LDCT images for diagnosis, and lung nodules larger than 10 mm. In total, 628 patients with SLNs (530 from Taipei, 42 from Hsinchu, and 56 from Sijhih CGH) were included in the CGH dataset (Appendix A). We also collected a publicly available American dataset (National Lung Screening Trial, NLST) containing LDCT data from The Cancer Imaging Archive (https://www.cancerimagingarchive.net/collection/nlst/, available on 4 April 2025) [24,25]. Among these patients, we excluded patients with the absence of detectable lesions or lung nodules larger than 10 mm, resulting in 7913 patients with SLN. To balance the sample size between the Asian and American datasets, we selected 600 patients from the 7913 SLN patients and ensured that the distributions of age, gender, and malignancy rates of the selected patients were consistent with those of the entire database of the NLST dataset (Appendix A). The inclusion criteria for both datasets were as follows: (a) the diameter of the lung nodule was less than 10 mm; (b) the malignancy of the nodule was confirmed by histology or longitudinal imaging follow-up; (c) the image quality of pre-treatment LDCT was sufficient. Age, gender, SLN volume, and SLN type were also recorded. The institutional review board of CGH approved this study (CGH IRB: CGH-P111079), and informed consent was waived because of the retrospective data collection. Data collection and all research methods in this study were performed in accordance with the Declaration of Helsinki and the regulations of CGH IRB.

### 2.2. Imaging Parameters of Low-Dose Computed Tomography

The LDCT data in the CGH dataset were obtained using 8 different CT scanners, including Aquilion 640 (Canon Medical Systems, Japan); Aquilion ONE (Canon Medical Systems, Japan); Aquilion Prime SP (Canon Medical Systems, Japan); Brilliance (Philips, Netherlands); Brilliance 64 (Philips, Netherlands); Ingenuity CT (Philips, Netherlands); Sensation 16 (Siemens, German); Vereos PET/CT (Philips, Netherlands). The scanning coverages are similar in all CT scanners, ranging from the thoracic inlet to the upper abdomen. CT images were reconstructed with a slice thickness of 1 to 5 mm. Pixel sizes ranged from 0.483 × 0.483 mm^2^ to 0.912 × 0.912 mm^2^. The matrix size for each slice was 512 × 512 with a 16-bit gray depth in Hounsfield units (HU). The peak tube voltage was 120 kVp, and the tube current ranged from 20 to 429 mA.

The LDCT data in the NLST dataset were acquired from 8 different CT scanners, including Aquilion (Canon Medical Systems, Japan); HiSpeed QX/i (GE HealthCare, Chicago, Illinois, USA); LightSpeed Plus (GE HealthCare, Chicago, Illinois, USA); LightSpeed QX/i (GE HealthCare, Chicago, Illinois, USA); LightSpeed 16 (GE HealthCare, Chicago, Illinois, USA); Mx8000 (Philips, Netherlands); Sensation 16 (Siemens, German); Volume Zoom (Siemens, German). The scanning coverage, matrix size, and gray depth for the NLST dataset were similar to those of the CGH dataset. CT images were reconstructed with a slice thickness of 2 to 5 mm. Pixel sizes ranged from 0.488 × 0.488 mm^2^ to 0.957 × 0.957 mm^2^. The peak tube voltage was 120 or 140 kVp, and the tube current ranged from 40 to 320 mA. Detailed information on CT scanners and imaging parameters for both datasets is listed in Appendix A.

### 2.3. Segmentation of Small Lung Nodules

The SLNs were delineated by two well-trained radiological technologists and verified by an experienced radiologist. The soft tissue window (window width: 350 HU, window level: 50 HU) and lung window (window width: 1500 HU, window level: −600 HU) on LDCT images were applied for lesion delineation. The soft tissue window was used to distinguish between SLNs and fluid components, such as pleural and pericardial effusions, and the lung window was applied to determine the border of SLNs. Several image preprocessing steps were employed to reduce the scanning variations. The spatial resolution of LDCT was adjusted to a voxel size of 1 × 1 × 3 mm^3^. The gray values of LDCT were adjusted to the lung window, followed by intensity normalization to a range from 0 to 255. The volume of SLNs was cropped into a matrix size of 40 × 40 × 13. Afterward, we separated the CGH and NLST datasets into the training (70% of SLNs) and test (30% of SLNs) sets, respectively.

### 2.4. Data Augmentation of the Training Sets

The images in the training set from the CGH dataset were augmented thrice by left-right flipping and rotating by 5 and 10 degrees. Contrarily, due to the imbalanced proportion of malignant and benign SLNs in the training set of the NLST dataset, the benign SLN images were augmented twice using left-right flipping and 5-degree rotation. The malignant SLN images in the NLST dataset were augmented six times by upside-down flipping, left-right flipping, and rotating by −10, −5, 5, and 10 degrees. All processes were conducted within the MATLAB R2023a environment (MathWorks Inc., Natick, MA, USA).

### 2.5. Network Architecture and Model Building

The two-channel U-Net was employed for the malignancy prediction in this study [26], which comprised an input layer, encoder blocks, a bottleneck block, decoder blocks, and a classifier using cross entropy as the loss function (Figure 1). In the first two encoder blocks, feature maps were processed through two pathways with different kernel sizes of convolutional layers: one with the kernel size of 3 × 3 × 1 and the other with the kernel size of 1 × 1 × 3. These maps were normalized via batch normalization, activated using a leaky ReLU, and downsampled with a maximum pooling layer (kernel: 5 × 5 × 5). The feature maps from the two pathways were concatenated in the bottleneck block, followed by a convolutional layer (kernel: 3 × 3 × 3). In the decoder blocks, transposed convolutional layers (kernel: 3 × 3 × 3) were applied, concatenating the two-pathway maps. Finally, fully connected layers, a softmax activation, and a final classifier were applied. The DL models were trained with the following hyperparameters: the stochastic gradient descent with momentum optimizer, an initial learning rate of 0.001, a drop factor of 0.5 for the learning rate, a batch size of 16, an L2 regularization of 0.01, and a momentum of 0.9. For TL, the model was initially pre-trained based on the source dataset (either the CGH or NLST dataset) and then fine-tuned by adjusting the weights of the final 8 layers in the two-channel U-Net based on the target dataset (the other dataset).

Overall, 5 models were built based on the same architecture in this study (Figure 1). **Models 1 (CGH model)** and **Model 2 (NLST model)** were trained solely based on the training set of the CGH and NLST datasets, respectively. **Model 3 (pooling model)** was built by pooling the training sets of the CGH and NLST datasets. **Model 4 (NLST model with TL)** was pre-trained based on the training set of the NLST dataset, followed by the TL on the training set of the CGH dataset. **Model 5 (CGH model with TL)** was pre-trained based on the training set of the CGH dataset, followed by the TL on the training set of the NLST dataset.

### 2.6. Assessment of Model Performance and Statistical Analysis

Differences in continuous and categorical variables between the CGH and NLST datasets were identified by two-sample t-tests and chi-square tests, respectively. The assessments of model performance were conducted based on the test sets. The sensitivity, specificity, and accuracy derived from the confusion matrix, as well as the area under the receiver operating characteristic curve (AUC), were used to assess the prediction performance of the models. To compare the performance among the five proposed models, the bootstrap method with 100 times resampling followed by a two-sample t-test with a Bonferroni correction was applied. To further explain how the networks predict malignancy probability for SLNs, we employed the analysis of gradient-weighted class activation mapping (Grad-CAM) in the prediction models [27]. Grad-CAM computed the gradient of the malignancy probability with respect to the feature maps of the final convolutional layer. We averaged the gradients to obtain importance weights, which were then used to produce a heatmap by performing a weighted combination of the feature maps. The resulting activation map highlighted the spatial regions that strongly influenced the model’s decision, allowing us to assess whether the model focused on clinically relevant regions/features.

## 3. Results

### 3.1. Demography of Study Cohorts

The study collected 669 lesions from 628 Asian patients at Cathay General Hospital (CGH), referred to as the CGH dataset. Additionally, we also collected an American da-taset by downloading 600 lesions from 600 patients in the National Lung Screening Trial (NLST), designated as the NLST dataset (Table 1). The age at diagnosis in the CGH da-taset (59.45 ± 13.07) was significantly younger (*p* < 0.001) than in the NLST dataset (61.48 ± 4.71). Females comprised a higher proportion (*p* < 0.001) in the CGH dataset (58.76%) compared to the NLST dataset (39.17%). No significant difference in the equivalent diameter and volume of SLNs was found between the two datasets. A higher proportion (*p* < 0.001) of malignancy was observed in the CGH dataset (47.09%) compared to the NLST dataset (28.83%). The distribution of SLN types (solid/partial solid/ground-glass opacity (GGO)) was significantly different (*p* < 0.001) between the CGH dataset (40.66%/30.79%/28.55%) and the NLST dataset (71.17%/10.00%/18.83%).

Additionally, we found that the malignancy probability of partial solid SLNs in the CGH dataset (49.03%) was significantly higher (*p* = 0.005) compared to the NLST dataset (28.33%, Appendix A). A similar finding was observed for GGO SLNs, with the malignancy probability being significantly higher (*p* < 0.001) in the CGH dataset (64.40%) compared to the NLST dataset (24.78%, Appendix A).

### 3.2. Reduced Model Performance in Different Datasets

Given these differences in SLN type distributions between the CGH and NLST datasets (Table 1), we further assessed whether the different distribution of SLN types between the CGH and NLST datasets leads to a reduction in model performance across populations. We separately developed prediction models for SLN malignancy based on the CGH and NLST datasets, referred to as **Model 1 (CGH model)** and **Model 2 (NLST model)**, and further applied these models to another dataset. Our results showed significant reductions in model performance when employing prediction models to different test populations (Figure 2). For instance, **Model 1 (CGH model)** showed significantly reduced performance (*p* < 0.001) in the test set of the NLST dataset (reductions in accuracy, sensitivity, and AUC of 18.4%, 97.9%, and 52.6%, respectively) compared to the test set of the CGH dataset (Figure 2A). **Model 2 (NLST model)** showed significantly reduced performance (*p* < 0.001) in the test set of the CGH dataset (reductions in accuracy, specificity, sensitivity, and AUC of 34.1%, 15.2%, 57.0%, and 43.6%, respectively) compared to the test set of the NLST dataset (Figure 2B).

### 3.3. Impact of Population Variation on Model Performance

A potential strategy to develop a prediction model with international application is training models using pooled datasets from different populations. In this study, we further investigated whether a prediction model (**Model 3**, **pooling model**) using both populations for training would improve the generalization of SLN malignancy prediction (Figure 2C). **Model 3** showed smaller differences in model performance (absolute differences in accuracy, sensitivity, and AUC of 0.16, 0.07, and 0.20, respectively) between the CGH and NLST dataset test sets compared to **Models 1** and **2**.

Appendix A shows the performance comparisons between **Models 1**, **2**, and **3** in the test sets of the CGH and NLST datasets. **Model 3** showed significantly poorer (*p* < 0.001) performance in the test set of the CGH dataset (reductions in accuracy, specificity, sensitivity, and AUC of 27.6%, 21.9%, 42.6%, and 32.0%, respectively) than in **Model 1**. The performance of **Model 3** is significantly lower (*p* < 0.001) in the test set of the NLST dataset (reductions in accuracy, sensitivity, and AUC of 10.2%, 29.1%, and 8.5%, respectively) than in **Model 2**. Accordingly, training a model by pooling the SLNs from different datasets reduced the performance for SLN malignancy prediction in both test sets.

### 3.4. Improvement of Model Performance Through TL

To overcome the challenge of the low performance induced by population variation and to facilitate international application, TL was applied to **Model 2 (NLST model)** to create **Model 4 (TL from NLST to CGH dataset)**, where the model parameters were fine-tuned in the final layers using the CGH dataset. Similarly, TL was applied to **Model 1 (CGH model)** to create **Model 5 (TL from CGH to NLST dataset)**, where the model parameters were fine-tuned in the final layers using the NLST dataset. Figure 3 shows the performance comparisons between models with and without TL. **Model 4 (TL from NLST to CGH dataset)** significantly enhanced (*p* < 0.001) the performance (increases in accuracy, F1 score, specificity, sensitivity, and AUC of 56.9%, 116.3%, 14.1%, 159.5%, and 83.0%, respectively) in the test set of CGH dataset compared to **Model 2** (Figure 3A). **Model 5 (TL from CGH to NLST dataset)** significantly enhanced (*p* < 0.001) the performance (increases in accuracy, F1 score, sensitivity, and AUC of 21.1%, 1520.0%, 3950.0%, and 95.7%, respectively) in the test set of the NLST dataset compared to **Model 1** (Figure 3B).

In the comparisons between the pooling model (**Model 3**) and models with TL, **Model 4** showed significantly better (*p* < 0.001) performance (increases in accuracy, F1 score, specificity, sensitivity, and AUC of 44.4%, 60.3%, 18.7%, 77.8%, and 47.0%, respectively) in the test set of CGH dataset compared to **Model 3** (Figure 3A). **Model 5** showed significantly better (*p* < 0.001) performance (increases in accuracy, F1 score, sensitivity, and AUC of 8.9%, 24.6%, 32.8%, and 4.7%, respectively) in the test set of the NLST dataset compared to **Model 3** (Figure 3B). Accordingly, TL could overcome the cross-population variation by adapting the parameters of the pre-trained model (built based on the source population) to the target population.

### 3.5. Training Time for the Proposed Models

The training time for the base models was 66.5 min for **Model 1**, 58.9 min for **Model 2**, and 130.5 min for **Model 3**. The training time for the TL was 21.4 min in **Model 4** and 13.8 min in **Model 5**.

### 3.6. Demonstrative Cases of Malignancy Prediction

To explore the interpretability of **Model 2 (NLST model)** and **Model 4 (TL from NLST to CGH dataset)**, the Grad-CAM was analyzed on two representative SLN cases from the CGH dataset (Figure 4). Case #1 was diagnosed with a benign solid SLN measuring 6.87 mm in diameter. In contrast to **Model 2**, which considered a more nonspecific area, including the ribs, **Model 4** focused on the lesion, associated vessels, and pleural retraction after TL. Case #2 was diagnosed with a malignant GGO SLN 1.56 mm in diameter. In contrast to **Model 2**, **Model 4** focused on the lesion, the vascular distribution, and the gap between SLN and pleura after TL. In summary, the model with TL could pay more attention to lesion-related areas and make a more accurate malignancy prediction.

## 4. Discussion

In this study, the CGH (Asian) and NLST (American) datasets showed differences in age at diagnosis, gender, and SLN types. We identified a reduced performance of specialized models (**Models 1** and **2**) in these distinct datasets (Figure 5A), potentially due to the population variations between datasets (Appendix A). Additionally, cross-population variations after pooling different datasets may also reduce the modeling performance in both datasets (Figure 5B). This study demonstrated that the models with TL showed enhanced performance compared to the initial models, hence improving the international application of prediction models for SLN malignancy across different populations (Figure 5C).

In this study, over 50% of cases in the CGH dataset were female. The CGH dataset also presented a younger age at diagnosis than the NLST dataset. The literature reports a young age and high proportion of female cases in the Asian cohort due to exposure to cooking oil fumes [11,28,29]. Compared to Americans, partial and GGO SLNs in Asians showed a distinctly higher malignancy risk, which has been revealed in the previous comparative study [15,16]. The variations in individual and SLN imaging characteristics across populations may mislead malignancy prediction when employing a model developed in different countries.

One potential solution to achieve satisfactory model performance across populations is to construct a generalized model by pooling datasets from multiple countries. However, our results showed that **Model 3**, based on a pooling dataset, could not effectively improve model performance (Figure 2), even though the imaging parameters (including image intensity and voxel size) had been adjusted to eliminate scanning variations [22]. Pooling different datasets with population inconsistencies can generate a high variance within the training set and interfere with the learning process of imaging patterns. Therefore, we considered that the poor performance of **Model 3** might be attributed to population variations. Instead of data pooling, an alternative approach to accommodate these inherent variations is required to facilitate the international application of the AI model.

This study identified the benefits of TL in enhancing the model performance in different datasets (Figure 3). First, the training times for **Models 1**, **2**, **3**, **4**, and **5** are 66.5, 58.9, 130.5, 21.4, and 13.8 min, respectively. The shorter training times observed in the TL models suggest that TL could adapt model parameters to population variations more efficiently than retraining the whole model. Second, compared to the specialized model, the TL models focused on SLN-associated features, including fibrosis, vascular distribution, and pleural retraction (Figure 4), all commonly used to evaluate nodule malignancy by radiologists and pulmonologists [30,31,32,33]. Furthermore, due to various image traits for SLNs, we also investigated the improvement of model performance by TL in different SLN types. Our sub-analysis showed that TL significantly improved performance across all three types in test sets of both the CGH and NLST datasets (Appendix A). We suggested that fine-tuning model parameters in TL could improve model performance, enabling the model to recognize the characteristics of different SLN types specific to the target population. Accordingly, TL could overcome population variations and facilitate the international application of malignancy prediction in SLN. Additionally, TL avoided data transfer across institutes and reduced the potential risk of data leakage. The TL technique offered a solution for developing international applications while ensuring compliance with privacy regulations in a short training time.

Limitations in this study were addressed as follows. First, because of the lack of clinical factors in the NLST dataset, we could only build the prediction model of SLN malignancy based on images. Although the proposed TL models could achieve an AUC above 0.90, additional benefits from including clinical risk factors, such as tuberculosis and chronic obstructive pulmonary disease [34], require further studies to confirm. Second, we reported superior performance of the employed two-channel U-Net compared to the ResNet and traditional U-Net for predicting malignancy of SLNs (Appendix A). Using advanced deep-learning architectures that are suitable for small lesions may further improve the malignancy prediction. Third, future studies, including more populations in addition to Asian and American datasets, might fully explore TL’s efficacy in facilitating the international application of SLN malignancy prediction.

This study observed the disparities in age at diagnosis, gender, SLN types, and malignancy rates between Asian and American datasets. Even though the images underwent harmonization, this study observed reduced performance when applying the prediction model to the independent external dataset. Finally, we suggest that TL could improve model performance and facilitate the international application of malignancy prediction models for SLNs.

## Figures and Tables

**Figure 1 diagnostics-15-01460-f001:**
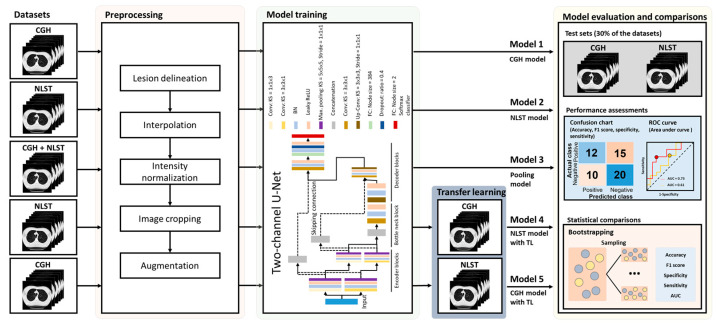
Diagram of study workflow. The LDCT images were preprocessed by resolution adjustment, lesion delineation, intensity normalization, image cropping, and data augmentation. The DL models were built based on processed training-set images using a two-channel U-Net. **Model 1** and **Model 2** were trained based on the training set of the CGH and NLST datasets, respectively. **Model 3** was built by pooling training sets of the CGH and NLST datasets. **Model 4** was trained based on the training set of the NLST dataset, followed by the transfer learning to the training set of the CGH dataset (pale blue block). **Model 5** was trained based on the training set of the CGH dataset, followed by the transfer learning to the training set of the NLST dataset (pale blue block). Assessments of the models were conducted by calculating the accuracy, sensitivity, specificity, and area under the receiver operating characteristics curve (AUC) based on the test sets of the CGH and NLST datasets, respectively. The statistical comparison between models was performed using the bootstrap method followed by a two-sample t-test. Conv: convolution layer, KS: kernel size, BN: batch normalization layer, Leaky ReLU: leaky Rectified Linear Unit, Concat: concatenate layer, up-Conv: transpose convolution layer, FC: fully connected layer.

**Figure 2 diagnostics-15-01460-f002:**
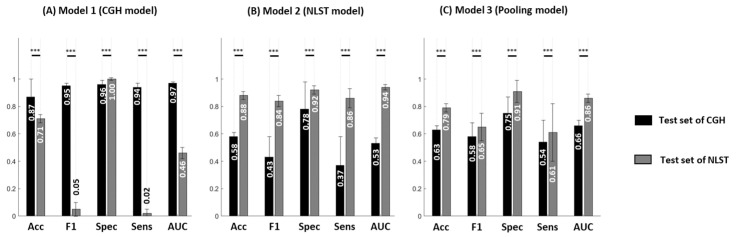
Comparisons of performance between the CGH and NLST datasets for two-channel U-Net models. (**A**) **Model 1** is the prediction model based on the CGH dataset. (**B**) **Model 2** is based on the NLST dataset. (**C**) **Model 3** is based on the pooling dataset. Each bar chart contains accuracy, F1 score, specificity, sensitivity, and AUC. Black and gray bars indicate the model performance in the test set of the CGH and NLST datasets, respectively. *** *p* < 0.001.

**Figure 3 diagnostics-15-01460-f003:**
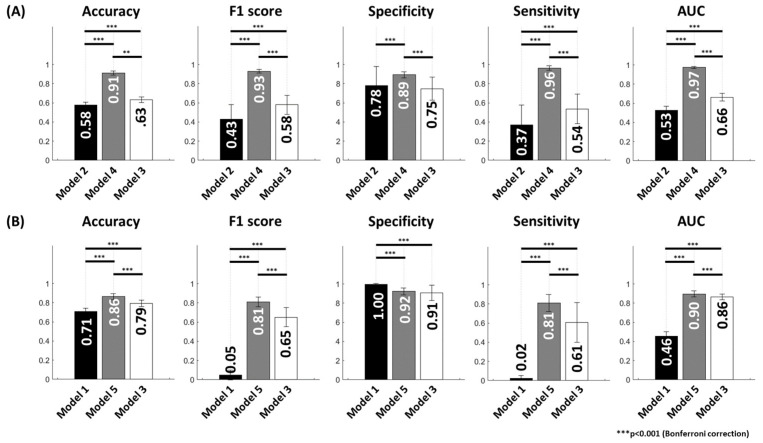
Performance comparisons in the target dataset among specialized models and TL models. (**A**) Accuracy, F1 scores, specificity, sensitivity, and AUC of **Model 2**, **Model 4**, and **Model 3** in the test set of the CGH dataset, respectively. (**B**) Accuracy, F1 scores, specificity, sensitivity, and AUC of **Model 1**, **Model 5**, and **Model 3** in the test set of the NLST dataset, respectively. ** *p* < 0.01, *** *p* < 0.001.

**Figure 4 diagnostics-15-01460-f004:**
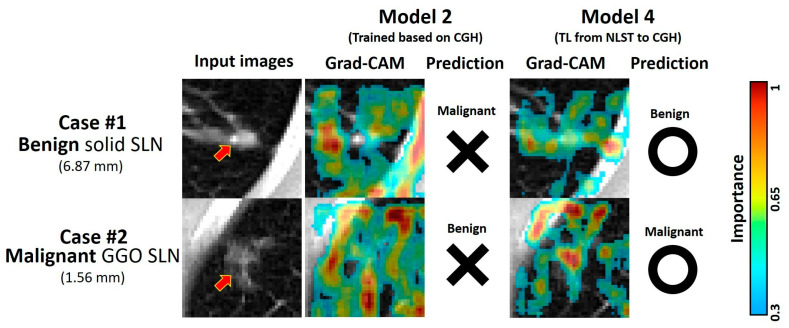
Grad-CAM and prediction for two representative SLN cases, which were mispredicted by **Model 2** but correctly predicted by **Model 4**. The Grad-CAM represents the importance of areas for malignancy prediction in the model, with red and blue colors representing high and low importance, respectively. The red arrows indicate the location of the SLNs. The circle and cross symbols indicate the correctness of the model prediction.

**Figure 5 diagnostics-15-01460-f005:**
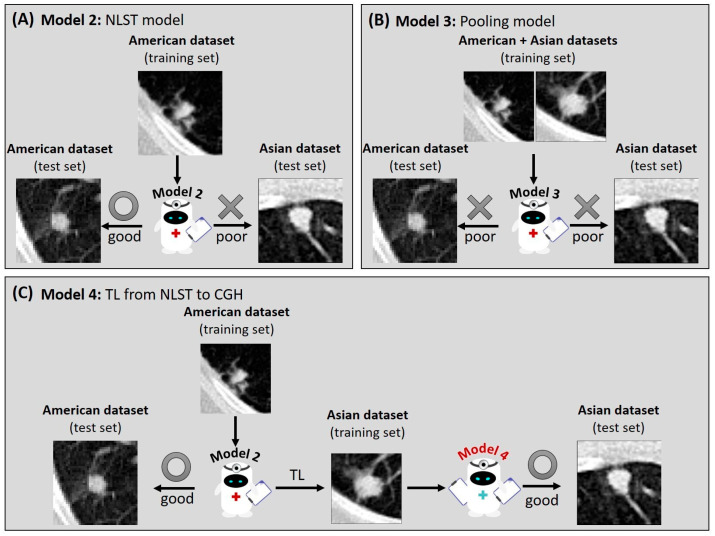
A diagram summarizing the benefits of TL. (**A**) **Model 2**, built based on the American dataset, performed poorly in the Asian dataset. (**B**) **Model 3**, trained by pooling American and Asian training sets, showed low performance in both datasets. (**C**) **Model 4**, built with TL from American to Asian datasets, overcame the cross-population variation and enhanced model performance to facilitate international application. The circle and cross symbols represent good and poor model performance in the test sets, respectively.

**Table 1 diagnostics-15-01460-t001:** Demography of study cohorts.

Characteristics	CGH Dataset	NLST Dataset	*p* Value
Patient number	628	600	
Age at diagnosis	59.45 ± 13.07	61.48 ± 4.71	<0.001 *
Gender (M/F)	259 (41.24%)/369 (58.76%)	365 (60.83%)/235 (39.17%)	<0.001 *
SLN number	669	600	
Pathology			<0.001 *
Benignness	354 (52.91%)	427 (71.17%)	
Malignancy	315 (47.09%)	173 (28.83%)	
Eq. Diameter (mm)	3.48 ± 1.86	4.37 ± 1.58	0.792
Volume (mm^3^)	557.96 ± 764.60	503.25 ± 619.38	0.453
SLN types			<0.001 *
Solid	272 (40.66%)	427 (71.17%)	
Partial solid	206 (30.79%)	60 (10.00%)	
GGO	191 (28.55%)	113 (18.83%)	

* *p* < 0.001.

## Data Availability

The raw data for the CGH dataset cannot be publicly available for ethical and legal reasons. However, researchers can submit inquiries for analyzed data to the corresponding authors by reasonable request. The NLST dataset can be accessed through the data portal of the Cancer Imaging Archive (https://www.cancerimagingarchive.net/collection/nlst/ available till 22 April 2025).

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
