# Peer review of "Enhanced Malignancy Prediction of Small Lung Nodules in Different Populations Using Transfer Learning on Low-Dose Computed Tomography"

_diagnostics, 2025, doi:10.3390/diagnostics15121460_

Round 1
Reviewer 1 Report
Comments and Suggestions for Authors
The paper considers the application of deep learning approach in the diagnosis of small lung nodules. The proposal model aims to take into account the demographic effect of populations. Although the medical application of the proposal approach could be valuable, I think that the paper has more explanation about the model since it is not defined clearly and needs more analyses to emphasize the novelty of the study. My suggestions are listed below:
1 - There is no any analysis based on the rejection of the source of differences between pair of means stated in Part 2.5. In order to clarify these sources, it would be good to conduct the Turkey pairwise test or simultaneous t-test with Bonferroni correction.
2 - In part 2.6, it is stated that the data is augmented by bootstrapping. Could you give more details about this augmentation please? Which approach do you use for bootstrap and why do you choose that specific approaches among alternative bootstrap strategies?
3 - Could you describe how you get TL model in Part 3.3 please? How do you pool the data? Please give more details. There is almost no details about the TL model.
4 - Could you present the results of F-measure in Figure 3 please as it can evaluate both types of errors simultaneously.
5 - Could you make Figure 5 more visible please?
Reviewer 2 Report
Comments and Suggestions for Authors
This manuscript describes the use of transfer learning to create a U-Net classification model that can work with international datasets for predicting malignancy in small lung nodules. The following questions and comments should be addressed during revision:
- Do models 1-5 share the same number of trainable parameters?
- The authors should also report on the training time for the base model and for TL.
- For TL, the authors should comment of the performance of the fine-tunned model on the parent data set (for example, model 4 on the NLST dataset) and compare that to the performance of the model before transfer learning to make sure that catastrophic forgetting has not occurred.
- How many data points were used in the TL for models 4 and 5?
- Interpretability of prediction/classification in important in medical, especially in diagnostic applications. The authors should comment on the interpretability of the proposed U-Net model so users can better under the reason behind the classifications made by the model.
Reviewer 3 Report
Comments and Suggestions for Authors
The manuscript entitled "International Application of Deep Learning Model for Predicting Malignancy in Small Lung Nodules" presents a methodologically sound and timely investigation into the use of transfer learning (TL) for improving malignancy prediction in small lung nodules (SLNs) from low-dose computed tomography (LDCT) scans. The authors leverage datasets from two geographically and demographically distinct populations (Taiwan and the United States), aiming to enhance cross-population model generalizability. The study addresses a critical challenge in the application of AI in medical imaging. However, there are several aspects that require further clarification.
1 - The current figures' resolution is rather low. Please fix this issue.
2 - Although the study utilizes benchmark datasets (CGH and NLST), the authors briefly mention that “The spatial resolution of LDCT was adjusted to a voxel size of 1×1×3 mm”. How the authors describe the details about data creation and enhancement?
3 - The dataset used is image data, the author says “). The age at diagnosis in the CGH dataset (59.45 ± 13.07) was significantly younger (p<0.001) than in the NLST dataset (61.48 ±4.71). “How the authors calculate the p-values?. The practical relevance of this statistical finding should be discussed?
4 - The title of sections should be accurate. “Reduction in model performance across population”
5 - The proposed model variants (Models 1 through 5) are not sufficiently described. It remains unclear how these models differ in terms of architecture, particularly in the number and types of layers.
6 - The explanation and interpretation of the Grad-CAM results in Figure 4 are inadequate. The authors should either provide more examples or discuss the limitations of Grad-CAM.
7 - The title of paper should be accurate or updated.
Reviewer 4 Report
Comments and Suggestions for Authors
The authors used transfer learning to improve malignancy prediction for small lung nodules using low-dose computed tomography across datasets from different countries.
Reviewer Comments:
1. The organisation and presentation of the manuscript are good.
2. Consider changing the word “International Applications” in the title. Many similar studies and applications already exist in this area. If you intend to use this word, please provide a valid justification.
3. In line 8, you mention "datasets from different countries." However, the manuscript only uses two datasets from Taiwan. Please provide evidence or revise the statement accordingly. Otherwise include more countries dataset for analysis.
4. Include a more detailed comparison with previous state-of-the-art methods. Discuss the most prominent and recent research approaches in the Introduction or Related Work (Add) section for better context.
5. authors using simple transfer learning for prediction. Why is this model is sufficient? How does this model improve performance? Please provide detailed explanations and comparisons with other models.
6. Include a comparative analysis of accuracy, sensitivity, and specificity using different state-of-the-art methods in the Results and Discussion sections.
Round 2
Reviewer 3 Report
Comments and Suggestions for Authors
I am satisfied with the author’s responses to my questions raised in my initial review. The revised manuscript is easier to follow based on feedback from the reviewers. I recommend that the revised manuscript be accepted for publication.
Reviewer 4 Report
Comments and Suggestions for Authors
Authors addressed all my comments.